# Developmental and Neurotoxicity of Acrylamide to Zebrafish

**DOI:** 10.3390/ijms22073518

**Published:** 2021-03-29

**Authors:** Jong-Su Park, Palas Samanta, Sangwoo Lee, Jieon Lee, Jae-Woo Cho, Hang-Suk Chun, Seokjoo Yoon, Woo-Keun Kim

**Affiliations:** 1Biosystem Research Group, Korea Institute of Toxicology, Daejeon 34111, Korea; jongsu.park@pitt.edu (J.-S.P.); samanta.palas2010@gmail.com (P.S.); sangwoo.lee@kitox.re.kr (S.L.); jieon.lee@kitox.re.kr (J.L.); hangsuk.chun@kitox.re.kr (H.-S.C.); 2Department of Environmental Science, Sukanta Mahavidyalaya, University of North Bengal, Dhupguri, West Bengal 735210, India; 3Toxic Pathology Research Group, Korea Institute of Toxicology, Daejeon 34111, Korea; cjwoo@kitox.re.kr; 4Molecular Toxicity Research Group, Korea Institute of Toxicology, Daejeon 34111, Korea; sjyoon@kitox.re.kr

**Keywords:** acrylamide, neurotoxicity, zebrafish, developmental toxicity, disease models, neurodevelopmental disorders

## Abstract

Acrylamide is a commonly used industrial chemical that is known to be neurotoxic to mammals. However, its developmental toxicity is rarely assessed in mammalian models because of the cost and complexity involved. We used zebrafish to assess the neurotoxicity, developmental and behavioral toxicity of acrylamide. At 6 h post fertilization, zebrafish embryos were exposed to four concentrations of acrylamide (10, 30, 100, or 300 mg/L) in a medium for 114 h. Acrylamide caused developmental toxicity characterized by yolk retention, scoliosis, swim bladder deficiency, and curvature of the body. Acrylamide also impaired locomotor activity, which was measured as swimming speed and distance traveled. In addition, treatment with 100 mg/L acrylamide shortened the width of the brain and spinal cord, indicating neuronal toxicity. In summary, acrylamide induces developmental toxicity and neurotoxicity in zebrafish. This can be used to study acrylamide neurotoxicity in a rapid and cost-efficient manner.

## 1. Introduction

Acrylamide is a water-soluble alkene primarily used to synthesize polyacrylamide for personal care products, and is also used in various chemical industries, wastewater treatment processes, chemical grouting, and soil conditioning [1,2]. Acrylamide is a common ingredient in plant-based foods, such as potato and grain products and in roasted coffee [3,4]. Its polymeric form is non-toxic but its monomeric form is highly toxic to rats and mice [5,6], with carcinogenic [7], teratogenic [8,9], and neurotoxic [1,5] effects. Human exposure to acrylamide results in neurotoxicity that is characterized by lethargy, skeletal muscle weakness, gait abnormalities, weight loss, ataxia, numbness of the extremities, and polyneuropathy [6,10]. Acrylamide neurotoxicity has been associated with central–peripheral distal axonopathy [11,12]. Molecular initiating events of acrylamide neurotoxicity include formation of adducts with sulfhydryl thiolate sites specifically involved in synaptic vesicle recycling in vesicle docking (synaptotagmin, synaptophysin, and syntaxin), vesicle priming (complexin 2), SNARE (SNAP Receptor) core dissolution (*N*-ethylmaleimide-sensitive factor), endocytosis (clathrin), neurotransmitter re-uptake (membrane dopamine transporter), and vesicular storage (vesicular monoamine transporter) at nerve terminals [13]. The developmental toxicity of acrylamide has been characterized in laboratory animals, but developmental neurotoxicity has not, highlighting the need for validated animal models of acrylamide-induced developmental neurotoxicity for the clinical management of patients affected by occupational exposure to acrylamide [14,15].

Zebrafish (*Danio rerio*) is a popular vertebrate model in developmental biology and has been used in neurotoxicity studies [16]. Its short generation time, high fecundity, and transparent body make it ideal for developmental toxicity assays [17], and it is suitable for the in vivo high-throughput screening of chemicals [18]. Additionally, zebrafish brain and central nervous system development occurs within 3 days post fertilization (dpf) [19]. Zebrafish share almost 70% DNA sequence homology with humans and possess similar neurotransmitters [20,21,22,23]. Hence, zebrafish are used as an intermediate model organism between cell-based assays and mammalian testing. Furthermore, chemical-induced malformations in zebrafish can easily be observed under a stereomicroscope because of their transparent bodies [24,25]. Acute acrylamide-induced neurotoxicity in adult zebrafish has been investigated previously with a 0.75 mM (51.31 mg/L) concentration [26,27], but the larval model is not well-established [18].

In this study, we exposed zebrafish embryos 6 h post fertilization (hpf) to acrylamide for 5 days to evaluate general toxic responses, developmental and behavioral outcomes, and neurotoxic and pathological effects. We aimed to verify the suitability of zebrafish larvae for acute acrylamide-induced neurotoxicity testing and to establish zebrafish as an alternative model.

## 2. Results

### 2.1. Acrylamide-Induced Developmental Defects

Zebrafish embryos exposed to acrylamide from 6 hpf, exhibited concentration- and time-dependent increases in mortality. An initial rise in mortality was observed after 72 h that continued throughout the exposure period. The LC_50_, LC_25_, and LC_10_ values at 120 hpf were 142.59, 54.77, and 30.85 mg/L, respectively.

Morphology, survival, hatching, and heart rate were measured in 24 h intervals (Figure 1). Various embryonic abnormalities were observed in acrylamide-treated fish from 24 to 120 hpf, including yolk retention, scoliosis, swim bladder deficiency, heart edema, and body curvature (Appendix A), with heart edema and yolk retention being the most pronounced. Exposure to 100 mg/L acrylamide shows 50–70% morphological malformation at 120 hpf. No mortality was observed in control zebrafish throughout the exposure period. Exposure to 300 mg/L acrylamide caused mortality at 72 hpf, and no fish survived at the 96-hpf measurement point (Figure 1A). The hatching rate was 100% in the control group, but this decreased gradually with increasing acrylamide concentrations to 53.3 ± 5.8% in the 300 mg/L acrylamide group (Figure 1B). Heartbeat decreased in a concentration-dependent manner (Figure 1C). Blood flow was reduced with increasing acrylamide concentrations throughout the exposure period. At 24 hpf, heart rate was 62.5 ± 0.5, 59.9 ± 0.4, 53.3 ± 0.2, 51.9 ± 0.7, and 46.0 ± 0.6 beats per minute in fish exposed to 0, 10, 30, 100, and 300 mg/L, respectively. At 48 hpf and 72 hpf, heart rate was 119.6 ± 4.1, 115.7 ± 2.1, 109.9 ± 2.5, 101.1 ± 2.9, and 89.7 ± 3.4 beats per minute and 147.6 ± 3.7, 138.9 ± 2.5, 129.6 ± 4.66, 121.5 ± 3.9, and 68.4 ± 9.6 beats per minute, respectively (n = 30). In the 300 mg/L acrylamide group, heart rate was much lower than in other groups, and by 80 hpf, all embryos had died.

### 2.2. Behavior

At 5 dpf, acrylamide treatment led to a decrease in total distance traveled and swimming speed in wild-type larvae (at 10 mg/L, 105.85 ± 8.39 cm and 0.19 ± 0.02 cm/s, respectively, n = 24; at 30 mg/L, 103.05 ± 6.93 cm and 0.18 ± 0.02 cm/s, respectively, n = 24). However, these values did not differ significantly from those recorded for the controls. Control group shows active swimming movement in whole plate, while 100 mg/L group shows swimming in the border of plate and circular movement, which indicated neuronal damaged related movement (Figure 2A, Appendix A). Treatment with 100 mg/L acrylamide induced a significant decrease in total distance traveled and swimming speed than in other exposure groups (78.47 ± 4.16 cm and 0.13 ± 0.01 cm/s, respectively; n = 24). In addition, larvae exposed to 100 mg/L acrylamide exhibited circular movement (Figure 2A–C). In control groups, zebrafish exhibited normal behavior in total distance traveled and swimming speed (128.56 ± 11.4 cm and 0.24 ± 0.03 cm/s, respectively; n = 24).

### 2.3. Neurotoxicity

Pan-neuronal cell fluorescence intensities of the transgenic *tg*(*elavl3:eGFP*) strain were analyzed to determine the presence of acrylamide-induced neurotoxicity by the width of axons in the brain and spinal cord (Figure 3A,A’,A”). At 48 hpf, brain width was unchanged in all treatment groups (181.21 ± 4.11, 184.26 ± 2.14, 188.13 ± 6.90, and 181.17 ± 4 µm for 0, 10, 30, and 100 mg/L, respectively; n = 10). At 72 hpf and 100 mg/L, brain width was slightly shorter than in controls (238.75 ± 4.04, 233.33 ± 6.03, 236.09 ± 5.34, and 220.17 ± 3.21 µm for 0, 10, 30, and 100 mg/L, respectively; n = 10; (Figure 3B,C). At 96 hpf, brain width was 259.77 ± 3.48 µm in controls and 252.98 ± 3.75, 253.23 ± 3.67, and 223.98 ± 7.94 µm in fish exposed to 10, 30, and 100 mg/L, respectively (n = 10). At 100 mg/L, brain width was consistently below control width until 120 hpf. At that time, brain width was 253.18 ± 4.53 in controls and 258.98 ± 5.84, 253.6 ± 4.45, and 227.57 ± 14.7 µm in fish exposed to 10, 30, and 100 mg/L, respectively (n = 10; Figure 3B,C). Overall, exposure to 100 mg/L acrylamide severely affected brain development in zebrafish.

Spinal cord neuronal width did not appear to be affected by any of the acrylamide exposures until 48 hpf (Figure 4A,B). However, measurements differed markedly at 72 hpf (58.55 ± 0.82 µm in controls and 61.33 ± 1.25, 61.26 ± 1.31, and 55.58 ± 1.38 µm in fish exposed to 10, 30, and 100 mg/L, respectively; *n* = 10). Spinal cord width in fish exposed to 100 mg/L was shorter than in controls (Figure 4B). At 96 hpf, spinal cord width decreased as acrylamide concentrations increased (60.34 ± 0.7 µm in controls and 60.11 ± 1, 58.53 ± 1.45, and 55.9 ± 0.76 µm in fish exposed to 10, 30, and 100 mg/L, respectively; n = 10). However, at 120 hpf, spinal cord width in fish exposed to 100 mg/L acrylamide (55 ± 2.09 µm) did not differ from spinal cord width in controls (58.53 ± 1.14 µm, *n* = 10; Figure 4A,B).

We did not observe defects in mature, myelinating oligodendrocytes or Schwann cells in the *tg*(*mbp:mGFP*) line (Appendix A and Appendix A) at 100 mg/L acrylamide. Similarly, the *tg*(*sox10:eGFP*) line spinal cord neurons showed normal numbers and morphology (Appendix A).

## 3. Discussion

The toxicity of acrylamide has been thoroughly demonstrated in rodent models using acute lethality and behavioral endpoints [4,28,29]. However, there is a lack of information on acrylamide-induced developmental toxicity in rodents because of the associated cost. Consequently, the use of zebrafish in studying developmental toxicity has increased steadily [30,31]. Acrylamide-induced behavioral, transcriptional, and proteome responses have been shown in zebrafish [18,22], and zebrafish have been used to evaluate the developmental neurotoxicity of other environmental toxicants [32]. Therefore, we aimed to establish zebrafish as a model of acrylamide-induced developmental neurotoxicity using standard toxicological endpoints.

Embryo mortality increased in a time- and concentration-dependent manner, consistent with previous reports. A 72 h LC_50_ value of 86.007 mg/L acrylamide has previously been calculated for adult zebrafish [22], similar to the 142.59 mg/L value we calculated for a 5-day exposure period. Rates of zebrafish lethality were comparable to rates observed in rodents [33,34,35]. Acrylamide also inhibited hatching with increasing concentration and duration of exposure. The reasons behind this could be due to the induction of developmental defects [36]. However, our results in 100 mg/L show that even malformation occurred, the hatching rate was not changed significantly. Additionally, these data suggest that developmental defects can cause delayed hatching, but are not the only factors affecting the hatching rate. Additionally, this demonstrates it’s the possible effects of acrylamide on the circulatory and nervous systems [24].

Acrylamide treatment caused a concentration-dependent increase in developmental malformations in the zebrafish embryos, which occurred within 24 h of acrylamide exposure, indicating possible teratogenic effects [36]. The early adverse physiological effects were observed in the yolk extension in the developing embryo, followed by pericardial yolk sac edema, swim bladder deficiency, and curvature of the body. Similar developmental abnormalities were reported in zebrafish embryos and larvae following psoralen exposure [17], indicating that organogenesis may be the developmental stage that is most sensitive to toxicants [36]. We found that the severity of these abnormalities increased with the acrylamide concentration and was accompanied by mortality [37]. Therefore, embryo mortality can be directly linked to the severity of developmental malformations. In addition, the reduced heart rate indicated cardiac dysfunction from pericardial edema [24,36] and included notable reductions in heartbeat. Pericardial edema formation collapses the yolk sphere, which prevents the heart from undergoing normal chamber formation [38]. Development of pericardial edema indicates structural cardiac abnormalities following acrylamide exposure [17,24]. Therefore, our findings indicate that acrylamide exposure led to the development of acute lethality and developmental toxicity accompanied by cardiac dysfunction in zebrafish embryos.

Generally, acrylamide developmental neurotoxicity in mammals is characterized by motor function defects [13]. We used basal locomotor activities, namely swimming speed and total distance traveled, as markers of neurotoxicity. The behavioral effects of acrylamide on zebrafish have not been studied extensively, but our data appear similar to those of a study that found a concentration-dependent decrease in swimming speed and total distance traveled [22]. Exposure to 100 mg/L acrylamide resulted in circular movements, indicating a movement disorder [38]. In order to rule out the consequences due to developmental defects, experiments were conducted only with embryos without developmental defects. Moreover, the reduction in swimming speed indicates toxicity to the developing nervous system [17]. We can, therefore, conclude that acrylamide induces damage to the brain and spinal cord neurons in zebrafish, leading to movement disorders not unlike those seen in humans [38].

Alterations in the width of brain and spinal cord neurons in *tg*(*elavl3:eGFP*) transgenic zebrafish indicated a toxic effect of acrylamide on the nervous system [17]. Zebrafish spinal cords are useful indicators because of their regeneration capacity and easily detected characteristics [37]. Additionally, the decline in brain and spinal cord widths at 72 hpf following exposure to 100 mg/L acrylamide indicated high toxicity. Accordingly, the *tg*(*elavl3:eGFP*) line is good for determining acrylamide neurotoxicity and associated neurotoxic modeling. Altogether, these findings indicate that acrylamide causes severe neurodevelopmental and teratogenic defects.

Previous mammal histological analyses confirmed whether neurological damage in rats was comparable to the outcomes observed in zebrafish, i.e., axonal degeneration and secondary demyelination [18,39]. Sciatic nerve axonal degeneration occurred in Wistar rats exposed to 30 mg/kg acrylamide [40,41]. Vacuolation either in the overlying layer or in striated arrangement from the Purkinje cell layer to the pial membrane indicates progressive cavitation and severe axonal degeneration [42,43]. Acrylamide caused Purkinje cell death, which was particularly prominent at the nerve terminal.

The transgenic *tg*(*elavl3:eGFP*) line expresses fluorescence in pan-neuronal cells, enabling the analysis of whole neurons. However, chemicals that only target specific regions of neurons or only specific cell types cannot be assessed using this transgenic strain, and normal fluorescence from adjacent cells may overlap. Therefore, additional laboratory techniques, such as immunohistochemistry, RNA-seq, and qPCR, are required to identify changes at the molecular or cellular level following exposure. In summary, acrylamide exerts developmental toxicity in zebrafish, which represent an alternative animal model for screening molecules for neurodevelopmental toxicity.

## 4. Materials and Methods

### 4.1. Animals

AB wild-type and *tg*(*elavl3:eGFP*) transgenic zebrafish were obtained from the Zebrafish Center for Disease Modeling, Republic of Korea, and maintained at 28 ± 0.5 °C with a 14/10 h light/dark cycle under standard conditions in a fish breeding circulatory system installed at the Korea Institute of Toxicology. Adult fish were fed *Artemia* twice a day. Embryos were produced by spawning, where two or three pairs of adult male and female zebrafish were placed overnight in mating tanks with shallow water, separated by a removable partition. After the partition was removed, fish were allowed to spawn and embryos were collected after 30 min. These were then washed with egg water (30 mg sea salt in 500 mL distilled water) and placed in a Petri dish. All experimental protocols and procedures were approved and conducted according to the guidelines and regulations of the Institutional Animal Care and Use Committee of the Korea Institute of Toxicology (approval number RS19006, July, 29th, 2019).

### 4.2. Acrylamide Experimental Procedure

Acrylamide (CAS 79-06-1, purity > 99%, M.W. 71.08) was purchased from Sigma-Aldrich (A9099; St. Louis, MO, USA). Exposure concentrations of 10, 30, 100, and 300 mg/L (0.14, 0.42, 1.4 and 4.2 mM, respectively) were selected from previous studies [18,36]. For zebrafish, test solutions were prepared on the day of the experiment in E3 medium (0.292 g NaCl, 0.013 g KCl, 0.044 g CaCl, and 0.081 g MgSO_4_ in 1 L) containing a 0.1% dimethyl sulfoxide (DMSO) solution. Control groups were exposed to 0.1% DMSO. For rats, test solutions were dissolved in distilled water for oral injection, and distilled water served as the vehicle control.

Healthy zebrafish embryos (both transgenic and AB) from the same developmental stage were selected using a microscope 6 hpf and placed in 12-well plates, with three replicates for each control and treatment group. Ten embryos were placed in each plates, dead and hatched embryos were counted and removed every day. The embryos were then immediately exposed to the selected acrylamide concentrations until 5 dpf. Media were replaced every 24 h with fresh 0.1% DMSO solution and the respective acrylamide concentrations. Transgenic zebrafish were used for assessing neurotoxicity and the AB wild-type strain was used for analyzing general developmental toxicity and behavior.

### 4.3. Lethality and Developmental Toxicity

The number of dead zebrafish per exposure group was recorded at each measurement time point (24, 48, 72, 96, and 120 h), and the half-maximal lethal concentration (LC_50_) was calculated. Lethality curves were established at each measurement time point. The hatching rate was also recorded. Heart rate was measured by recording heartbeat for 15 s and extrapolating to beats per minute under tricaine (ethyl 3-aminobenzoate methane sulfonate salt; Sigma-Aldrich (St.Louis, MO, USA); pH 7 adjusted by Tris to pH 9) treatment to anesthetic embryos. This measurement was repeated for three replicates per larva and the mean heart rate was used as the individual heart rate. For phenotypic or morphological analyses, zebrafish were observed and photographed at each measurement time point to check for malformations. The survival, hatching, heartbeat, and malformation rates were calculated manually.

### 4.4. Behavior

Total distance traveled and swimming speed were analyzed in AB larvae at 5 dpf. To exclude consequences due to morphological malformations, we use no morphological defect larvae in behavior test. Twenty-four larvae from each group were introduced into 24-well plates (one larva per well) containing 1 mL E3 medium and placed in a DanioVision recording system equipped with EthoVision XT11 version software (Noldus, Wageningen, The Netherlands). Larvae were then acclimatized for 10 min in the dark followed by tracking for 10 min in light. All measurements were performed in the afternoon between 2:00 and 5:00 PM.

### 4.5. Neurotoxicity

*tg*(*elavl3:eGFP*) zebrafish from the control and acrylamide groups were anesthetized using tricaine and mounted on glass slides, then examined and photographed under a fluorescence microscope (Zeiss AXIO Imager Z2, Oberkochen, Germany). Brain (mid-region of midbrain) and spinal cord (behind the swim bladder) widths were examined and calculated using ImageJ software (National Institutes of Health, Bethesda, MD, USA).

### 4.6. Statistical Analysis

All data are represented as means ± standard error of the mean. The results were analyzed by GraphPad Prism version 5 (San Diego, CA, USA). For the normality and homogeneity of each variance, the Shapiro–Wilk test and the Levene’s test were used, respectively. Data follow a normal distribution. One-way analysis of variance followed by the Bonferroni post hoc test or Student’s *t*-test were used to identify differences between groups, with *p* ≤ 0.01 considered statistically significant.

## Figures and Tables

**Figure 1 ijms-22-03518-f001:**
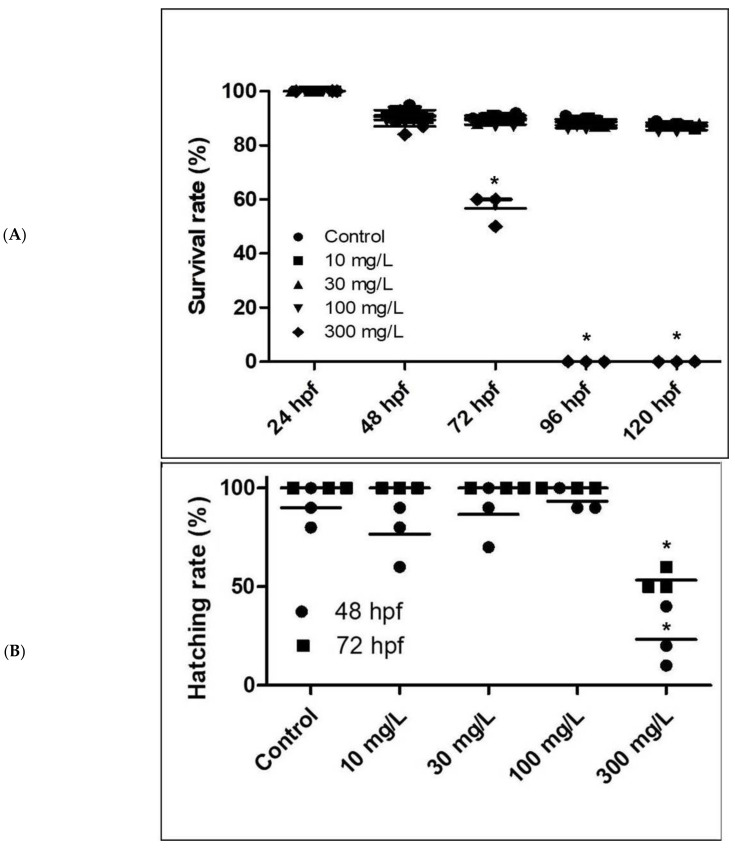
Acrylamide exposure negatively affects survival (**A**), hatching (**B**), and heart rate (**C**). *n* = 30 for each dose, respectively. * *p* < 0.01. Error bars represent the standard error of the mean.

**Figure 2 ijms-22-03518-f002:**
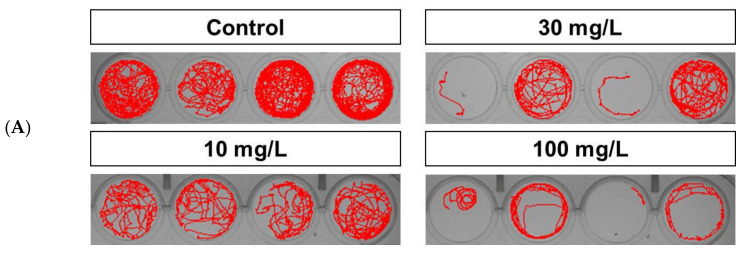
Behavioral effects following a 5-day exposure to acrylamide. (**A**). Acrylamide-treated wild-type zebrafish larvae exhibited impaired locomotor activity (circular movement and decreased swimming speed and distance traveled) that was concentration-dependent. (**B**). Distance traveled. (**C**). Swimming speed. * *p* < 0.01. Error bars represent the standard error of the mean.

**Figure 3 ijms-22-03518-f003:**
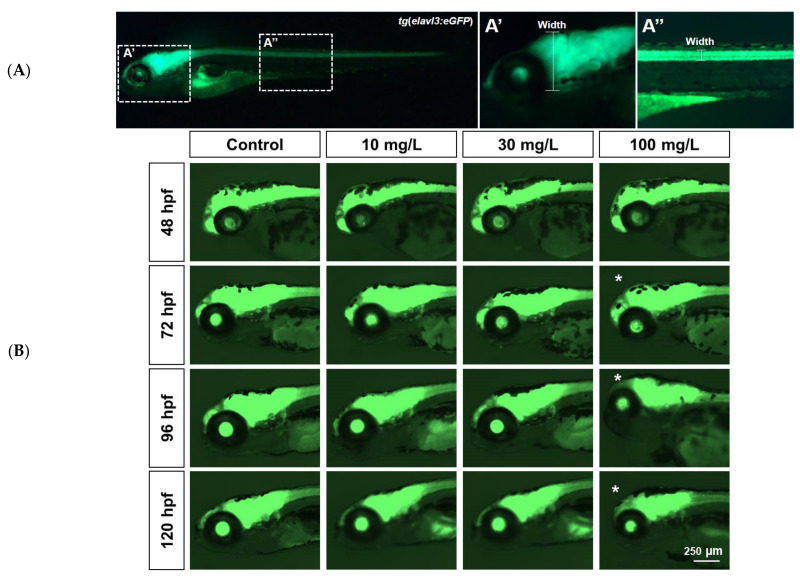
Acrylamide-induced neurotoxicity in transgenic *tg*(*elavl3:eGFP*) zebrafish. (**A**–A”). Pan-neuronal cells were used to analyze neurotoxicity. Brain width (A’) and spinal cord neuron width (A”) were measured. (**B**). Brain neurotoxicity. From 72 hpf onwards, zebrafish larvae treated with 100 mg/L acrylamide exhibited shorter brain width than controls. White asterisks denote severely short brain width. (**C**). Brain width. Brain width was shorter at the 72–120 hpf measurement time points in fish exposed to 100 mg/L acrylamide (*n* = 10). * *p* < 0.01. Error bars represent the standard error of the mean.

**Figure 4 ijms-22-03518-f004:**
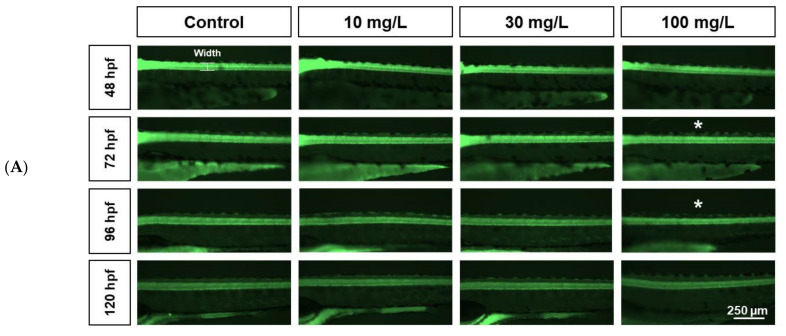
Comparison of spinal cord neuronal width between controls and acrylamide-treated zebrafish. (**A**). Spinal cord neuron width was similar between groups until 48 hpf. Between 72 and 96 hpf, fish treated with 100 mg/L acrylamide exhibited shorter spinal cord width (white asterisks). (**B**). Spinal cord neuronal width (*n* = 10). * *p* < 0.01. Error bars represent the standard error of the mean.

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
