# Peer review of "Developmental and Neurotoxicity of Acrylamide to Zebrafish"

_ijms, 2021, doi:10.3390/ijms22073518_

Round 1

Reviewer 1 Report

Developmental and neurotoxicity of acrylamide to zebrafish

Park et al., 2021

General:

The manuscript is well written and the premiss is interesting. My concern is that the majority of the impacts detailed in the manuscript (behavioural and morphological impacts) have not been fully determined to be independent of potential malformation/ developmental toxicity. Impacts on the heart rate at early stage could be an indicator as observed at 30mg/l or lower but other impacts potentially could be related to malformation of the zebrafish.

The authors need to be able to delineate between impacts caused by morphological malformations and the neurological toxicity. Currently the discussion and the results do not take the malformations observed and how that could relate to behavioural impacts or morphological neural endpoints observed.  Before this could be used as an alternative to studying acrylamide toxicity in rats this area of overlap needs to be investigated. The comparison to the rat also required more detail to allow stronger justification for the use of the zebrafish.

Specific:

Abstract

None at the current moment but potentially alter to include changes to the results and discussion.

Introduction

Line 49: Please provide reference for the animal studies highlighted in this passage.

Line 63: The authors reference a paper of adult exposure; to allow comparison to this published work and others quoted in this manuscript can the authors put in the relevant concentration of exposure; in this case 0.75mM (51.31mg/l).

General: The authors reference toxicity exposure to acrylamide, relationship of these concentrations to either environmental exposure or to long term exposure will help relate the dose ranges used in this manuscript.

Results

General: The reduced hatching rate in spencer et at 2018 was found at 100mg/l but not in this paper, how do they respond to the difference? Also, malformations at 100mg/l - how does that effect interpretation of the data gathered in relation to neurotoxicity?

General: Behavioural impacts could not determine a significant alteration below 100mg/l but with known malformation of the larvae at 100mg/l then how significant are these impacts and separate from general toxicity or caused by the malformation of the zebrafish?

Section 2.1:

The authors describe the impacts of the acrylamide treatment on the developing embryo-larvae but they do not detail the timing or frequency of these impacts on the larvae, the number of larvae treated in each group or present that data in a format that can be interrogated. Without this data it is not possible to compare this to the other published data and is required in this manuscript to be able to interpret the other endpoints investigated.

Understanding the malformation present within the 100mg/l treatment group is critical the interpreting the other endpoints recorded including hatching rate, heart rate, motility and structure of the neural tissue. In figure S1 there is clear malformation of the jaw and alteration of the forebrain of the zebrafish larvae at 5dpf at 100mg/l. Malformation frequencies are not recorded in the manuscript even though the ref paper by Spencer et al., 2018 details between 60-20% malformations at 100mg/l. This needs to be integrated and discussed by the authors.

Figure 1: Remove the break on the y axis and bring x axis labels horizontal. Additionally, see if there are other dose concentration markers which could be used in a black and white figure to help delineate between the different treatments. N number to be included in figure legend.

Figure 2: N number required in this figure and all related figures

Line 117: without the determination that there has not been a reduction in length of the zebrafish at 72hpf or greater at 100mg/l then reduction in other body structures cannot be considered to be independent of potentially general toxicity.

Figure 3 B: there is potentially malformation of the 100mg/l fish at both 96 and 120hpf (and even at 72hpf but that is more difficult to definitively determine from one image). Again, with understanding the background level of malformation then it is not possible to place the results in relation to the toxicity to the zebrafish.

Section 2.4: The authors discuss the histological impact in rats but do not detail the concentration that this is observed in or any other toxicity data related to the rat study. As the manuscript is using the comparison to the rat for justification for the use of the ZF as an alternative. I would expect a higher level of detail about the rat data and any related toxicity observed or not observed along with behavioural data on the rat study to allow greater comparison between the two species epically if the premise of this study is to support the use of the zebrafish over the rat in assessment of acrylamide toxicity.

Discussion

General: The paper would benefit from additional histology on the zebrafish neural to allow comparison if possible.

Line 183: the authors state possible teratogenic effects and reference Spencer et al 2018, the paper in the discussion states “developmental abnormalities in the zebrafish embryos, which leads to developmental arrest and delayed hatching, especially in the highest concentration” possible teratogenic needs to be changed to developmental defects.

Line 203: the authors need to prove the behavioural impacts observed are independent of the toxic impacts observed.

Materials and methods:

Line 255: please state the Molecular weight of the compound as all dosing information is in mg/l and allows easier conversion to mM concentration for comparison with other studies or compounds.

Line 262: State how many embryos were placed in each well of the 12-well plate, if more than one embryo per plate please state if embryos which had become necrotic were removed from the well once recorded.

Line 270: why those concentrations of oral application of acrylamide chosen? The ref 36. (Lopachin) describes and experiment at either 175 or 400mg/kg/day while this experiment is conducted at 3-25mg/kg(/day).

Line 272: state the N number of each group as it is not possible to determine the N number from the information in the manuscript.

Line 275: was the heart rate recorded please state what media it was recorded in and what system used to review these recordings.

Section 4.3: The assessment of both heart rate and morphology was taken at multiple timepoints during development, normally this is done using a low level of anaesthetic at about >72hpf due to larval movement. None is mentioned in the manuscript, either include how this was done without anaesthetics or what anaesthetic was used and at what concentration. Also the methods state these were conducted at 24-120hpf but only data for 24-72hpf is presented in figure 1 for 24-72hpf.

Line 285: Please state the lighting conditions of the chamber during assessment as it is not stated if the assessment was carried out in the dark or light or with a stimulation with light pulses.

Line 311-314:  The statical analysis needs to state if a test for normality was conducted before the ANOVA assessment was conducted. Depending on the size of each group (which is not stated) then a small group would not necessarily have a normal distribution.

Reviewer 2 Report

The authors show in this manuscript an interesting work about the neurodevelopmental neurotoxicity of acrylamide in zebrafish. Consequently, this work should be suitable for publication in International Journal of Molecular Sciences. However, major changes have to take into account:

This manuscript shows interesting data about the effect of acrylamide in exposed embryos of zebra fish. The results in zebra fish are very interesting. However, my principal concern is about the results in rats. The rats exposed to acrylamide are adult and therefore is not possible to validate the neurodevelopmental toxicity in zebra fish with the neurotoxic effects in adult rats, because the zebra fish exposed are embryos. Furthermore, the authors do not indicate what is the dose of the data of exposed rats showed in Results. Did the rats exposed to the different doses show similar pathological changes?, what are the findings for each dose of acrylamide in rats?, why did the authors use several doses in rats?, why did the authors use adult rats to validate neurodevelopmental toxicity in embryos of zebra fish?. All these questions must be properly answered and therefore the section 2.4 and the section Discussion have to be rewritten including the answers.

Reviewer 3 Report

This manuscript shows how the zebrafish is a good model to study the toxic effects of acrylamide.

Below, I write comments / suggestions on each section of the manuscript that I hope will help the authors to improve it.

Introduction:

In the introduction, there is less reference to some recent work, for example, which shows that exposure to acrylamide induces skeletal developmental toxicity in zebrafish (Zhu, F., Wang, J., Jiao, J., & Zhang, Y. (2021). Exposure to acrylamide induces skeletal developmental toxicity in zebrafish and rat embryos. Environmental pollution 271, 116395. https://doi.org/10.1016/j.envpol.2020.116395) or Faria et al, in which the zebrafish is characterized as a model of acrylamide acute neurotoxicity (Faria, M., Valls, A., Prats, E., Bedrossiantz, J., Orozco, M., Porta, J. M., Gómez-Oliván, L. M., & Raldúa, D. (2019). Further characterization of the zebrafish model of acrylamide acute neurotoxicity: gait abnormalities and oxidative stress. Scientific reports, 9(1), 7075. https://doi.org/10.1038/s41598-019-43647-z), or Huang and cowoekers, who studied the effect of acrylamide on heart development in zebrafish (Huang, M., Zhu, F., Jiao, J., Wang, J., & Zhang, Y. (2019). Exposure to acrylamide disrupts cardiomyocyte interactions during ventricular morphogenesis in zebrafish embryos. The Science of the total environment, 656, 1337–1345. https://doi.org/10.1016/j.scitotenv.2018.11.216). Another reference that is missed is "Krishnan, M., & Kang, S. C. (2019). Vitexin inhibits acrylamide-induced neuroinflammation and improves behavioral changes in zebrafish larvae. Neurotoxicology and teratology, 74, 106811. https://doi.org/10.1016/j.ntt.2019.106811" in which the authors have found some results similar to those shown in this work, for example, how acrylamide produces swimming impairments.

Results:

About all the graphs presented: it would be better to present data in a manner that more transparently shows the sample size and distribution. Swarm plots, for example, are greatly preferable to bar graphs. (See PLoS Biol. 2015; 13:e1002128. doi: 10.1371/journal.pbio.1002128).

In figure 1A, if possible, it would be better, in addition to different symbols for each group, using different colors to facilitate the interpretation of the graph.

In figure 3A, A 'and A' 'are missing the scale bar.

In the supplementary data, I suggest to provide at least one video showing a zebrafish control vs treated with acrylamide to see the behavioural.

The spinal cord is relatively homogeneous but not 100%, and its width varies from rostral to caudal. How have the authors chosen the area in which to measure the width of the spinal cord?

Throughout the manuscript reference is made to "axon width" but both, by the values and by the images, it seems that what they mean is spinal cord width. They should correct this or clarify better what they refer to.

Material and methods

Have the authors performed a normality analysis to decide whether to use a parametric test or a nonparametric test?

The number of animals is mentioned throughout the manuscript, but the work would gain clarity by adding a table that shows the number of animals used for each experiment.

Discussion:

The manuscript could be improved if, in the discussion, it is highlighted which are the new data that this work contributes to the previously existing ones.

In general

The experiments that were carried out are well designed, well explained and the results are reliable. However, in my opinion, the work lacks novelty since, in general, it is a confirmation of previously published data. Something that could make a difference is to delve into the results and try to explain them, for example, what causes the reduction of the brain? It could be that acrylamide increases neuronal death (a tunnel experiment would demonstrate this) or, perhaps, acrylamide reduces neurogenesis (an experiment with BrdU, PCNA labeling or similar could shed light on this question).

Round 2

Reviewer 2 Report

On the one hand, I do not agree with the authors answer, it is not possible to compare the neurodevelopmental toxicity of treated embryos exposed to acrylamide with the toxic effects in adult rats treated with this chemical and therefore the data of the neurotoxic effects observed in adults rats after exposure to acrylamide can not use to validate the zebra fish as animal model of neurodevelopmental toxicity.

On the second hand, the authors do not answer to the questions related to the experiment with rats that I asked: what is the dose of the data of exposed rats showed in Results?, did the rats exposed to the different doses show similar pathological changes?, what are the findings for each dose of acrylamide in rats?, why did the authors use several doses in rats?.

Taking into account these reasons the manuscript could be published if the authors remove the experiment with adults rats because it is not useful to validate neurodevelopmental toxicity in zebra fish.

Reviewer 3 Report

Now, the manuscript is significantly better, important references have been added and in the video of the supplementary material it is evident how higher doses of acrylamide affect the behavior of the zebrafish.

The “scientific” part of the manuscript could be considered of sufficient quality to be published. However, the appearance of the manuscript could be improved. I do not agree with the authors when they said that making graph by swarm plots when there are several X-axis data makes more difficult to understand. Swarm plots are always clearer because they show all the observed points, it clearly shows the number of samples, the value of each sample and the variability. Also, in my opinion, whenever an image is shown it should go with a scale bar because it allows the reader a better interpretation of the image.

Regarding the normality test necessary to carry out the parametric statistical analysis: the authors have answered that they have carried it out, but I think they should add it to the manuscript in section “4.7. Statistical analysis”, indicating which normality test they have used and whether the data follow a normal distribution or not.

In conclusion, the results presented here, in my opinion, are reliable. I think that the manuscript could be improved, however, I consider that the fundamental role of the reviewer is to check that the experiments are well planned, with the correct controls and statistics well performed. The format of the manuscript is not the main role of the reviewer; therefore I will not dwell on this.

Round 3

Reviewer 2 Report

Accept in present form